# Neurocognitive Correlates of Clinical Decision Making: A Pilot Study Using Electroencephalography

**DOI:** 10.3390/brainsci13121661

**Published:** 2023-11-30

**Authors:** Serkan Toy, Somayeh B. Shafiei, Sahin Ozsoy, James Abernathy, Eda Bozdemir, Kristofer K. Rau, Deborah A. Schwengel

**Affiliations:** 1Departments of Basic Science Education & Health Systems and Implementation Science, Virginia Tech Carilion School of Medicine, Roanoke, VA 24016, USA; serkantoy@vt.edu; 2Intelligent Cancer Care Laboratory, Department of Urology, Roswell Park Comprehensive Cancer Center, Buffalo, NY 14263, USA; somayeh.besharatshafiei@roswellpark.org; 3BioSoftPro, LLC, Kensington, MD 20895, USA; sahin@biosoftpro.com; 4Department of Anesthesiology and Critical Care Medicine, The Johns Hopkins University, 1800 Orleans Street, Baltimore, MD 21287, USA; jaberna4@jhmi.edu; 5Department of Pathology, Yale School of Medicine, New Haven, CT 06520, USA; eda.bozdemir@yale.edu; 6Department of Basic Science Education, Virginia Tech Carilion School of Medicine, Roanoke, VA 24016, USA; krisrau@vt.edu

**Keywords:** medical education, clinical reasoning, decision making, electroencephalogram, neurocognitive evidence

## Abstract

The development of sound clinical reasoning, while essential for optimal patient care, can be quite an elusive process. Researchers typically rely on a self-report or observational measures to study decision making, but clinicians’ reasoning processes may not be apparent to themselves or outside observers. This study explored electroencephalography (EEG) to examine neurocognitive correlates of clinical decision making during a simulated American Board of Anesthesiology-style standardized oral exam. Eight novice anesthesiology residents and eight fellows who had recently passed their board exams were included in the study. Measures included EEG recordings from each participant, demographic information, self-reported cognitive load, and observed performance. To examine neurocognitive correlates of clinical decision making, power spectral density (PSD) and functional connectivity between pairs of EEG channels were analyzed. Although both groups reported similar cognitive load (*p* = 0.840), fellows outperformed novices based on performance scores (*p* < 0.001). PSD showed no significant differences between the groups. Several coherence features showed significant differences between fellows and residents, mostly related to the channels within the frontal, between the frontal and parietal, and between the frontal and temporal areas. The functional connectivity patterns found in this study could provide some clues for future hypothesis-driven studies in examining the underlying cognitive processes that lead to better clinical reasoning.

## 1. Introduction

Developing sound medical decision making is essential for optimal patient care and safety. The decision-making process in medicine is complex and dynamic and requires concerted reasoning efforts for integrating a substantial fund of medical knowledge into often incomplete and unfolding case-specific information [1]. Healthcare professionals frequently deal with ambiguity, time constraints, and various limitations while making crucial decisions about patient care, which can have serious consequences [2]. Experience and deliberate medical practice, along with critical thinking and reflection, are thought to be the essential factors in developing this skill [3]. 

However, medical decision-making processes can be quite elusive as clinicians are not always aware of the psychological underpinnings of their own decision-making process, which may not be readily apparent to an independent observer either [1]. Thus, it is challenging to teach and assess the quality of the reasoning process for clinical decision making. In addition, a disparity can arise between the quality of the reasoning process and the eventual patient outcome, which could potentially be influenced by a host of other variables. Some theoretical frameworks exist to elucidate the dynamics of this process. For instance, according to cognitive theories of dual processing, individuals commonly use two distinct forms of reasoning: type 1, characterized by automatic, unconscious, and implicit processes, and type 2, marked by controlled, conscious, and explicit cognitive functions [4]. 

It is assumed that with experience and proficiency, reflective and more effortful decision making evolves into more intuitive judgments [5]. Cognitive researchers suggest that newly acquired information is encoded and stored in an interconnected network of knowledge structures [6]. This network subsequently facilitates a streamlined information search and retrieval during the reasoning process [7]. The faster mode of reasoning often relies on this interconnected web of medical knowledge and prior experience with diverse case-specific information in the form of schemas and illness scripts [8]. 

This is, in fact, congruent with the neural efficiency hypothesis that suggests individuals with higher cognitive functioning show lower brain activation than those with lower intelligence while working on the same cognitive tasks [9,10]. Expanding on this hypothesis, Neubauer and Fink’s (2009) literature review concluded that neural efficiency is not only related to intelligence but may also be a function of knowledge and experience pertaining to the task in question [11]. Cognitive shortcuts like heuristics and pattern recognition developed through proficiency can offload some of the burden on working memory to allow multiple factors to be considered during clinical decision making. 

Working memory serves two primary functions: short-term storage and cognitive control [12,13]. Neuroimaging studies have shown that working memory functions are processed by the frontal cortex. The pre-frontal cortex (PFC) areas show consistent modality-specific (verbal versus spatial versus object information) activation patterns, which generally appear to mediate rehearsal processes for verbal and spatial information [13]. 

Neuroscientists have used functional neuroimaging as a continuous measure of neurocognitive activities since the 1990s [14]. Currently, this technology encompasses a range of methods, including but not limited to functional magnetic resonance imaging (fMRI), functional near-infrared spectroscopy (fNIRS), and electroencephalography (EEG). Both fMRI and fNIRS rely on the hemodynamic response for capturing neurocognitive processes. fMRI necessitates research participants to remain stationary within the scanner during a cognitive task engagement, which could be impractical in educational and operational settings. This could also limit the ecological validity of such studies [15,16]. Currently, fNIRS and EEG have wireless, relatively non-invasive solutions that can be used to measure neurocognitive processes within realistic learning environments [14,15,17]. 

A recent scoping review revealed an emerging area of research using neuroimaging evidence to investigate various clinical skills in health profession education research [18]. Most of these studies focused on psychomotor skill development, yet a few provided preliminary information on the neurocognitive aspects of clinical reasoning and decision making, supporting the theoretical assumptions. For example, Rotgans and colleagues (2019) hypothesized that working to diagnose unfamiliar cases would trigger “type 2” thinking, which could be observed as an activation of the prefrontal cortex. Using functional near-infrared spectroscopy (fNIRS), researchers were able to show that only unfamiliar cases (but not previously trained ones) invoked a significant level of activation in the prefrontal cortex while the medical students diagnosed chest X-rays [19]. 

Hruska and colleagues (2016), using fMRI, showed that reading clinical vignettes activated several regions of the brain including occipital, prefrontal, parietal, and temporal regions. The results also indicated that novices relied on their working memory more than experts [20]. Regarding the hemispheric activation patterns, novices and experts exhibited no discernible differences while engaged in the clinical decision-making process for straightforward cases. However, in challenging cases, novices showed greater activation on the left hemisphere of the prefrontal cortex (PFC), whereas experts had more activation on the right PFC [21]. Another group found that incorrect answers were associated with significantly higher PFC activation than correct answers while responding to multiple-choice questions previously utilized in licensing examinations [22]. 

These findings offer a degree of support to the dual processing theory. However, medical decision making is a multifaceted process that likely involves multiple areas of the brain. There needs to be more work in this vein to better understand the neurocognitive dynamics of the clinical decision-making process. Most studies used fMRI technology to measure neurocognitive processes during clinical decision making. Although this technology has a high spatial resolution, it often has a low temporal resolution. Additionally, it can be restrictive due to the expensive and complex equipment set-up. Subjects also need to remain relatively still in a confined space. EEG, on the other hand, offers better temporal resolution and is a non-invasive, relatively low-cost portable solution to conduct research in ecologically valid operational settings [14,15]. The lack of prior research using EEG to study clinical reasoning requires exploratory pilot studies to serve as a foundation for larger hypothesis-driven studies. 

The functional significance of EEG oscillations in different frequency bands (e.g., Theta, Alpha, Beta, and Gamma) has been explored in various laboratory settings with limited experimental tasks like Stroop or Go/NoGo tasks [23,24,25]. Theta activation has been shown in tasks requiring controlled attention, information retrieval processes, cognitive workload, and cognitive control [26,27,28,29]. A study using a coherence analysis illustrated Theta oscillations may play a role in coordinating functional interactions in the frontal–parietal neural networks (between the executive and attentional control regions) during decision-making tasks [30]. Increased Alpha oscillations typically indicate the resting cortical tissue [31]. However, certain tasks require inhibiting task-irrelevant memory, and this selective memory access may also be associated with increased Alpha-band activity [31,32]. Beta waves play a role in the active concentration [33], and frontal Beta, in particular, seems to control the contents of the short-term memory [34]. The Gamma band is associated with attention and complex information processing, mainly in localized neural networks [31,35,36]. Although we now have a general understanding of what these neural oscillations might mean for cognitive functioning [37], there are no well-established findings for clinical reasoning tasks in realistic settings. 

Many medical specialty boards in the United States and around the world employ rigorous high-stakes oral examinations to ascertain a physician’s eligibility for certification as a diplomate in their respective specialty. This procedure relies on trained physicians as examiners and independent raters, cases, and rubrics that have undergone reliability and validity testing. We simulated an abbreviated version of the standardized oral exam (SOE) of the American Board of Anesthesiology (ABA) to explore the differences in the brain dynamics of experienced examinees and novices. 

Although this was an exploratory study, our working hypotheses were that both groups would exhibit high levels of cognitive engagement and that experienced trainees would show more efficient brain activation patterns marked by better cognitive control and application of their medical knowledge. We also expected that multiple brain regions would be involved during clinical reasoning. Thus, a coherence analysis may be a better measure of the underlying complex cognitive processes than power spectral density alone (activation patterns at each electrode location), as it provides estimates of functional interactions between activated brain areas [38,39].

## 2. Materials and Methods

The local Institutional Review Board (IRB file number: IRB00200456) approved this study.

### 2.1. Participants and Procedures

We recruited junior anesthesiology residents and graduating fellows at a large quaternary care academic medical center on the East Coast of the United States. All participating fellows recently passed their American Board of Anesthesiology (ABA) written and standardized oral board exam (SOE) and were recently certified anesthesiologists. The junior anesthesiology residents comprised Clinical Anesthesiology Year 1 (CA1) residents, all of whom were unfamiliar with the SOE process and its format. We recruited anesthesiology trainees affiliated with our institution who were healthy adults with no known cognitive, behavioral, or psychological issues. There were no other exclusion criteria. 

Participants provided verbal consent, completed pretest materials and surveys, and were fitted with a wireless EEG headset following the study protocol. Baseline EEG recordings for each participant were collected over a 3-min period with the participant’s eyes open. Then, each study participant received an 18-min SOE based on the format used by the ABA. Finally, participants completed post-test measures. The entire data collection process took place between August and October 2019. 

### 2.2. American Board of Anesthesiology-Style Standardized Oral Board Exam

The SOE lasted a total of 18 min. After receiving the stem, the participants were given 3 min to read and prepare to answer questions about the case. The initial preparatory phase was followed by 12 min when an experienced ABA examiner asked a series of predetermined and scripted questions based on the clinical vignette. In the final 3 min, the candidates were presented with a different case vignette with additional scripted questions. The examiner evaluated each candidate in real-time using a rubric similar to the ABA’s, while an independent anesthesiologist reviewed and scored recorded sessions using the same rubric. See Appendix A for the mock SOE clinical case, additional questions, and the assessment form. 

The ABA SOE assesses clinical reasoning, ability to adjust to unanticipated, dynamic clinical situations, organization, and communication of information. The additional topic (last 3 min) requires more adaptive thinking and responding in real-time since there is no preparation time.

### 2.3. EEG Data Acquisition

In this study, the participants wore the 14-channel wireless EMOTIV EPOC+ headset. The channel names and locations are AF3, F7, F3, FC5, T7, P7, O1, O2, P8, T8, FC6, F4, F8, and AF4. See Figure 1. 

All channels in this headset are mastoid-referenced. The headset is equipped with a single 2048-Hz-rate analog-to-digital converter (ADC) that sequentially samples the channels and provides a filtered data rate of 256 samples per second. The data are 14 bits, and the least significant bit represents 0.51 microvolts. During the down-sampling process, which converts the sampling rate from 2048 Hz to 256 Hz, data are filtered using a digital 5th-order sinc filter and two notch filters at 50 Hz and 60 Hz to remove noise. The resulting output is a clean signal with a sampling rate of 256 Hz in the frequency range of 0.2–45 Hz (Epoc User Manual, https://emotiv.gitbook.io/epoc-user-manual/ accessed on 21 November 2023). This EEG headset uses Ag/AgCl electrodes, and the electrode–scalp connection is achieved using wet felts soaked in a saline solution. The quality of the connection was ensured through impedance measurement. EEG data were captured and recorded in an XDF file for an analysis [40]. 

EEG meta-data and event markers were incorporated into the data to document specific details of EEG recordings and to denote the start/stop times of various segments within the exam. As an illustration, instances like “eyes open” and “eyes closed” are clearly marked with these labels. The initial 3-min period allocated for reading and thinking about the clinical case stem is denoted as “reading”, the subsequent 12 min designated for questioning on this case are identified as “answer 1” and the final 3 min of questioning related to a new case vignette are labeled as “answer 2”. 

### 2.4. Measures

In addition to EEG recordings, we also collected self-reported cognitive load ratings using the NASA-Task Load Index (TLX). NASA-TLX includes 6 items on a 20-point visual analog scale (0 = “very low” to 20 = “very high”) tapping into mental demand, physical demand, temporal demand, performance, effort, and frustration. Typically, the combined score from all 6 subscales is used to measure the total cognitive load (0–120) [41,42].

Observed task performance scores were collected using an analytical checklist to serve as a decision-making performance score (Appendix A). We also collected demographic information in the pretest survey. 

### 2.5. Observational Assessment of Decision-Making Performance

The assessment form used a 3-point scale (Yes = 2, Maybe = 1, and No = 0) to generate quantitative scores. We calculated a weighted percent for each participant by dividing the total weighted score by the total possible score. 

### 2.6. Data Analysis

The EEG data analysis was carried out using MATLAB© (R2018A, provided by Mathworks Inc. 1 Apple Hill Drive, Natick, MA, USA), EEGLAB [43], and Brainstorm [44], which is available for download under the general public license (http://neuroimage.usc.edu/brainstorm accessed on 15 March 2020). The initial handling of the XDF file was carried out using EEGLAB. This constituted merging several XDF files into one single entity, when necessary, along with validating and rectifying channel locations within the XDF file. Then, all data were imported into Brainstorm, and a new study database was generated. Artifact marking and rejection were performed using Brainstorm functions. A custom MATLAB code was generated to split the data into 1 s intervals, eliminate any intervals with artifacts, and calculate QEEG variables for each participant relative to baseline eyes-open amplitudes. This process helps control for individual differences in EEG signals and helps isolate task-related changes in EEG values. The percentage of the epochs discarded was around 10%, and they were due to the instances when there was excessive head/muscle movement. The EEG recordings were mostly clean during the actual task. The artifact removal was performed semi-automatically, using an EEGLAB function. The analysis was purely fast Fourier transform (FFT) with Kaiser windowing of the epochs to prevent spectral leakage.

Our analysis framework is based on the observation that neural networks generate oscillations in different frequency ranges depending on the cognitive task. These frequency ranges include delta waves (1–3 Hz), Theta waves (4–7 Hz), Alpha waves (8–12 Hz), Beta waves (13–30 Hz), and Gamma waves (>30 Hz) [45]. To examine neurocognitive correlates of clinical decision making, two types of features were extracted from the recorded EEG signals: (1) power spectral density (PSD) features extracted by short-time Fourier transform with a Kaiser window, and (2) using a coherence analysis, functional connectivity between pairs of channels was extracted (a total of 91 features for each individual task and frequency band). Baseline correction was applied by extracting the eyes-open baseline values from all the values in the epoch before the analyses [46]. Before analyses, all features were normalized using the ‘Min-Max Scaling’ process. This process scales the values to a range between 0 and 1, where the minimum value of the feature vector is mapped to 0, and the maximum value is mapped to 1.

As mentioned, the abridged SOE included the following tasks, each with a predetermined time allowance: 3 min for initial reading and thinking about the clinical case stem, 12 min for questioning about this case, and 3 more minutes for discussing a new case vignette. The recordings were labeled according to this structure to reflect the main cognitive tasks and analyzed separately as reading, answer 1, and answer 2. 

All self-report and observational data were transformed into percentage scores (ranging from 0 to 1 in decimal increments) by dividing each raw score by its possible maximum score. This rescaling facilitated meaningful comparisons between measures. Intra-class correlation coefficients (ICCs) were calculated to evaluate the inter-rater reliability for weighted mock applied exam performance scores between two independent raters. We used independent samples *t*-tests to compare group differences on EEG estimates, self-reported cognitive load scores, and observed task performance. We also constructed two linear models for examining the extent to which relevant EEG features explain the variance in the (i) performance scores and (ii) cognitive load (TLX) scores. All analyses were performed in MATLAB or Statistical Package for the Social Sciences (IBM SPSS Statistics for Mac, Version 27.0; IBM Corp, Armonk, NY, USA), with a significance level set at *p* < 0.05. 

## 3. Results

### 3.1. Demographics

A total of sixteen participants (eight novice, first-year anesthesiology residents; eight experienced, graduating fellows) completed all study materials and were included in the study. The mean age for novice trainees was 29.13 (SD = 3.10); for experienced trainees, it was 32.75 (SD = 3.28); *p* = 0.039 for age difference. There was a similar gender distribution in both groups (novice group, four females/four males; experienced group, five females/three males). All of those in the experienced group reported participating in three or more mock applied exam practices, whereas none in the novice group reported participating in any applied exam practices. All participating fellows and experienced trainees passed their American Board of Anesthesiology (ABA) applied board examination within the last 6 months at the time of this study, and novices indicated that they would sit for this exam in 2 to 3 years. Only one participant, who was an experienced trainee, was left-handed. All the other participants were right-handed. 

### 3.2. Self-Report and Observational Measures

There was no difference between the novice (mean = 0.66, SD = 0.09) and experienced trainees (mean = 0.63, SD = 0.17) in terms of self-report cognitive load scores (*p* = 0.840). However, experienced trainees (mean = 93%, SD = 5%) significantly outperformed novices (mean = 54%, SD = 13%) based on the weighted performance scores (*p* < 0.001).

Inter-rater reliability was measured by using intraclass correlation coefficients (ICCs). A two-way random effects model for absolute agreement showed a high degree of agreement in the raters’ scores, 0.886, with a 95% confidence interval from 0.683 to 0.960 (F15,15 = 9.330, *p* < 0.001).

### 3.3. Change in Individual EEG Features across Expertise Levels

There were no significant differences between the novice and experienced groups for power spectral density. The two groups’ coherence features showed significant differences for all three exam phases. Table 1 shows the significant differences in coherence between brain areas of experienced and novice groups during the applied exam in all three tasks: initial reading phase, answering for case 1, and answering for case 2.

While reading the clinical vignette for case 1, fellows (i.e., experienced trainees) showed significantly higher fronto-parietal coherence in the Theta and Gamma bands. Novices exhibited greater functional connectivity in the Theta and Alpha bands among frontal channels and in the Gamma band between frontal and occipital channels. See Figure 2.

During the questioning of the first clinical case, it was observed that experienced trainees (fellows) had greater coherence in the Theta, Beta, and Gamma bands within the frontal area. However, novices showed higher functional interaction between the channels in the temporal and parietal regions in the Gamma band (Figure 3).

During the questioning of spontaneously presented case 2, significant features in the Beta and Gamma bands are higher for novices. See Figure 4. 

### 3.4. Development of Evaluation Models for Performance and Cognitive Load Using EEG Features

We entered the EEG features that significantly correlated (Pearson correlation; *p*-value < 0.05) with performance in a linear model using the L1 penalized (penalized Lasso method) to select variables and estimate coefficients simultaneously. The Lasso feature selection technique was used because we were dealing with a large number of EEG features and small sample size, and needed to pick only important features in explaining output variance without overfitting. To tune the key parameter (lambda) of the model, we used the five-fold cross validation technique and identified the lambda value that produced the lowest values of mean squared error (MSE). None of the features individually contributed significantly to explaining the variance in performance, but collectively, the model with the features included in Table 2 explained 94% of the variance in the performance scores (R^2^ = 0.94). 

For the cognitive load scores (TLX), again, features significantly correlated with NASA-TLX were used in a linear model with the penalized Lasso method to select variables and estimate coefficients. Theta coherence in the right prefrontal cortex (PFC), between channels F4 and F8, had a significant positive relationship with the cognitive load score (*p* = 0.026), and interhemispheric Theta coherence in the PFC had a significant negative relationship with the cognitive load scores; see Table 3. With three EEG features, the model shown in Table 3 explained 81% of the variance in the cognitive load scores (R^2^ = 0.81). 

## 4. Discussion

This study explored the use of EEG during a simulated standardized board-style oral exam to provide insights into the underlying brain dynamics of clinical reasoning by examining the patterns of neural activity shown by two distinct groups of individuals. Graduating fellows had successfully passed their board exams within the past 6 months at the time of this study. They also reported participating in several practice oral exams and focused studying in preparation for the American Board of Anesthesiology applied exam. On the other hand, novice anesthesiology residents had just started their program and had no experience with the board-style oral exam. We expected that fellows would have a good grasp of the oral exam format and a stronger fund of medical knowledge compared to the junior residents. Thus, the finding of fellows outperforming residents is no surprise. 

However, both groups reporting similar cognitive load is somewhat unexpected. In our previous work, we found that experts outperformed novices in a simulated endotracheal intubation task and reported significantly lower cognitive load [47]. The current finding highlights that the relationship between cognitive load and task performance depends on the specific nature of the task. While endotracheal intubation is a psychomotor skill that can be performed with nominal working memory engagement by experts, in this study, clinical reasoning required substantial mental effort from both groups. With similar mental effort, fellows outperformed junior residents. The high cognitive load did not appear to hinder their performance. This observation could further confirm that fellows possessed more robust medical knowledge and could utilize their cognitive resources effectively during clinical reasoning. Additionally, the fellows indicated that the exam format used in the study evoked some stress due to their recent experience with the oral exam, but they managed to perform well under stress. Future studies should examine the differential effect of stress and cognitive load on clinical reasoning performance in different individuals.

The EEG results revealed that the notable distinctions between the experienced group (fellows) and novices (junior residents) predominantly revolved around the inter- and intrahemispheric coherence of channels within the frontal area, as well as the coherence between frontal and parietal areas, frontal and temporal areas, frontal and occipital areas, and temporal and occipital areas. These neural activities and communication patterns within and between brain regions in these distinct groups can provide some clues for examining the underlying cognitive processes that lead to better clinical reasoning. 

In this study, the frontal region plays an important role in differentiating clinical reasoning skills. The frontal lobe of the brain has been associated with several cognitive processes, such as attention, working memory, and decision making. The temporal cortex plays a role in interpreting auditory input, language comprehension and production of speech processing, and processing emotions. The parietal area of the brain is involved in attentional control, numerical processing, and somatosensory information integration. The occipital cortex plays a role in visual perception. Considering the significant features (coherence between channels) and the frequency range, results can be interpreted separately for each study phase. 

### 4.1. Reading Phase

Fellows showed significantly higher fronto-parietal coherence in the Theta and Gamma bands. It has been suggested that executive functions are not solely dependent on activation in the prefrontal cortex; rather, they involve a distributed fronto-parietal network. This may point to the recruitment of distinct brain networks responsible for the central executive processes [48]. Numerous studies reported anterior (frontal) and posterior (temporal/parietal) Theta phase synchronization for different working memory tasks [49,50]. This network also plays a crucial role in governing the goal-directed attentional selection process, which is an important component of perception [51,52,53]. A study using EEG phase clustering analyses and fMRI responses further demonstrated that Theta phase synchronization was implicated in central executive circuits, working memory, and cognitive action sequences. This was particularly evident in the right frontal and left parietal regions of the brain, aligning with our results [54].

However, during Theta and Alpha bands, functional interactions for channels within the frontal area and in the Gamma band between channels from frontal and occipital areas are higher for novices. This may indicate that experts could use their working memory more efficiently without the need for a high level of functional interaction in the frontal region (related to working memory) to formulate their thoughts in anticipation of a clinical decision-making task. However, novices were not able to use their working memory autonomously and needed a high level of functional interaction within channels within the frontal area and between channels from the frontal and occipital areas. 

In the Theta band, experienced individuals showed lower coherence than novices between ‘FC5’ and ‘FC6’. Theta is often associated with memory processes and navigation. The higher coherence in novices might suggest greater mental effort or engagement during the reading phase, potentially reflecting less efficiency in information processing. For Alpha and Gamma bands, similar patterns were observed. Alpha is linked with relaxation and calmness, whereas Gamma is associated with higher cognitive functions and information processing. The differences could suggest that experienced individuals processed information differently (as seen with Gamma coherence variations).

### 4.2. Answer 1 Phase

The coherence of channels within the frontal area is significantly higher for experts in the Theta, Beta, and Gamma bands. It may indicate that experts have activated relevant medical knowledge and are involved in integrating environmental information (including case-specific questions and prompts) while making clinical decisions. Theta oscillations are particularly important for decision making in the frontal cortex, and their coherence reflects the cognitive control [30]. 

However, functional interaction between channels in the temporal and parietal regions during the Gamma band was higher for novices than for experts. It may be related to the nature of this phase being a question-and-answer format. Coherence in the temporal and parietal cortices is critical for language comprehension and production of language. Higher coherence between these regions for novices may indicate that they devoted more attentional resources to understanding the case-specific questions and formulating answers, while experts’ brains processed those types of information more efficiently with less functional interaction. 

The experienced group showed higher Theta, Beta, and Gamma coherence for certain channels (notably ‘F7’ and ‘F8’) during this phase. This finding could imply that during the problem-solving tasks, the experienced individuals’ brains exhibited more synchronized activity in regions necessary for these cognitive tasks. Since Beta is associated with active concentration and Gamma with complex information processing, higher coherence in experienced individuals might reflect more efficient problem-solving strategies or neural pathways developed by their experience. Table 4 outlines the overall findings of this study with summary discussion points. 

### 4.3. Answer 2 Phase

All significantly different features are higher for novices than experts. Results of answer 1 and answer 2 may indicate that those phases are more challenging than the reading phase, and functional interactions are higher for novices while trying to make decisions. These results may also indicate that throughout the exam, questions become more straightforward for experts while those become more challenging for novices because questions become more detailed and based on their previous responses. Therefore, significant features during Beta and Gamma bands are higher for novices, showing a higher demand on cognitive resources toward decision making. However, this pattern of neural dynamics did not appear efficient when interpreted along with their low-performance scores. 

In this phase of the exam, we found more varied differences, especially in the Beta and Gamma bands. Novices generally showed higher coherence in these bands between several channel pairs. This increased coherence might indicate that novices were exerting more cognitive effort, possibly because they were less familiar with the task at hand. The experienced group’s lower coherence suggests they might have developed more streamlined cognitive processes for the task, requiring less inter-regional communication.

### 4.4. EEG Features for Predicting Clinical Reasoning Performance and Cognitive Load

Finally, the linear models indicated that certain EEG features could be used to predict performance and self-reported cognitive load levels in a clinical reasoning task. Models developed using the Lasso feature selection technique showed that extracted features can evaluate performance and cognitive load with reasonable precision. The Lasso technique selected EEG features that are effective in performance and cognitive load evaluation via shrinkage, not a classical criterion (*p*-value < 0.05). Selected features were not statistically significant (i.e., op < 0.05) for performance, but all together were able to explain 94% of the variation in the performance scores. Coherence measures between various electrode pairs, mainly in the Gamma (and some in the Alpha and Beta) frequency bands, were important in performance prediction. These bands are often associated with cognitive functions, with Gamma, for instance, being involved in higher processing tasks, attention, and perception. The coherence means that the level of synchronized brain activities between different brain regions during the exam played an important role in performance, which can be an indicator of network connectivity utilized during complex tasks like decision making and problem solving. 

Additionally, Beta power (specifically at channel F7) played a role in performance, which could be related to active, busy, or anxious thinking and the active concentration required for decision making. Beta waves are most commonly linked to a state of wakefulness, alertness, and concentrated cognitive activity. These waves are prominent during logical thinking and conscious thought, which aligns with the requirements of a high-risk decision-making scenario, like the board-style oral exam, which demands complex decisions based on a given clinical vignette.

Despite the high R^2^ value, none of the individual EEG features were significant on their own, based on traditional *p*-value criteria. This suggests that no single EEG measure could be identified as a ‘key’ to high performance. Instead, successful decision-making and problem-solving performance is likely a complex, multifaceted process involving the interplay of various neural synchronizations and activations. It may indicate that decision making is not merely about increased activity or connectivity in one region but a symphony of interactions across various networks. A larger sample size is required to explore the validity of these findings in a broader population.

The high R^2^ value of the NASA-TLX prediction model indicates a strong fit between the model and the actual data, suggesting that these EEG features are robust indicators of cognitive load in this application and using this dataset. The significant predictors (with *p*-values indicating statistical significance) imply that the synchronization of brain activity, especially in the Theta band within and between hemispheres, is crucial in how participants experience cognitive load.

We found that increased Theta coherence between channels F4 and F8 in the right prefrontal cortex (PFC) is significantly associated with higher cognitive load scores. Theta waves are often linked to tasks requiring memory, navigation, and attention. Higher coherence could reflect more synchronized cognitive processes needed to manage or adapt to high-demand tasks. 

Conversely, increased interhemispheric Theta coherence (across the left and right PFC) correlated with lower cognitive load scores. This might indicate that efficient communication and balance between the two hemispheres facilitate task processing, reducing the perceived load.

## 5. Conclusions

This pilot study provides evidence that EEG could serve as a viable measure to complement traditional assessment modalities, offering valuable insights into the neurocognitive engagement of learners during clinical reasoning tasks. The functional connectivity patterns found in this study could provide clues for future hypothesis-driven studies examining the underlying cognitive processes that lead to better clinical reasoning. The results highlighted the complexity of brain dynamics as physicians make decisions under pressure. However, more research with larger samples is needed to understand how these EEG patterns might translate into actionable training or assessment strategies. 

## Figures and Tables

**Figure 1 brainsci-13-01661-f001:**
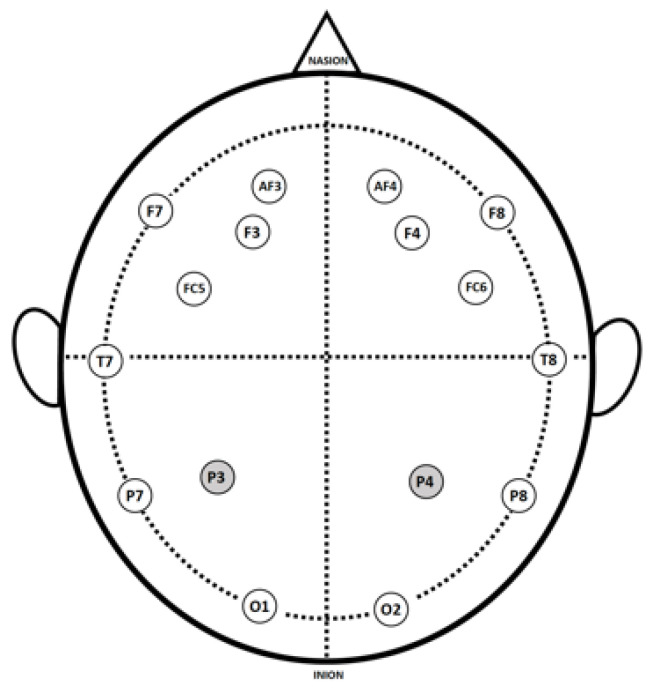
EEG channel locations; P3 and P4 denote reference electrodes.

**Figure 2 brainsci-13-01661-f002:**
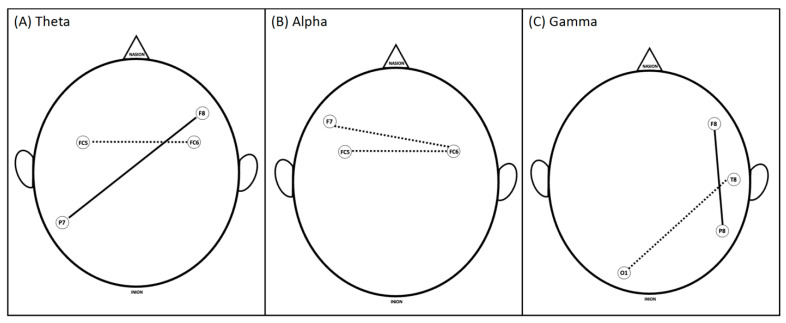
Significant differences in coherence between brain areas of experienced and novice groups during the initial reading phase (reading and thinking about the clinical vignette) for (**A**) Theta, (**B**) Alpha, and (**C**) Gamma frequency bands. Solid lines show where the experienced group had significantly higher coherence than the novices; dashed lines show where novices had significantly higher coherence.

**Figure 3 brainsci-13-01661-f003:**
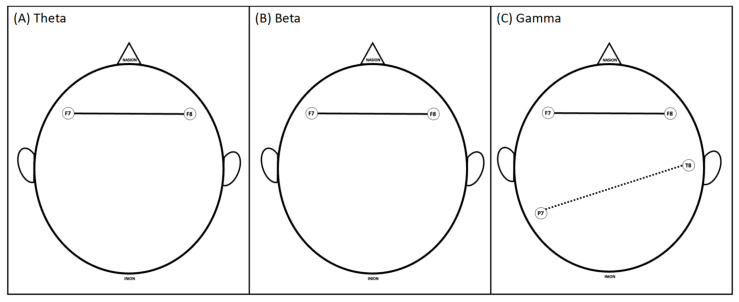
Significant differences in coherence between brain areas of experienced and novice groups during the applied exam for the first case (labeled as answer 1; question-and-answer format similar to the standardized oral board exam) for (**A**) Theta, (**B**) Beta, and (**C**) Gamma frequency bands. Solid lines show where the experienced group had significantly higher coherence than the novices; dashed lines show where novices had significantly higher coherence.

**Figure 4 brainsci-13-01661-f004:**
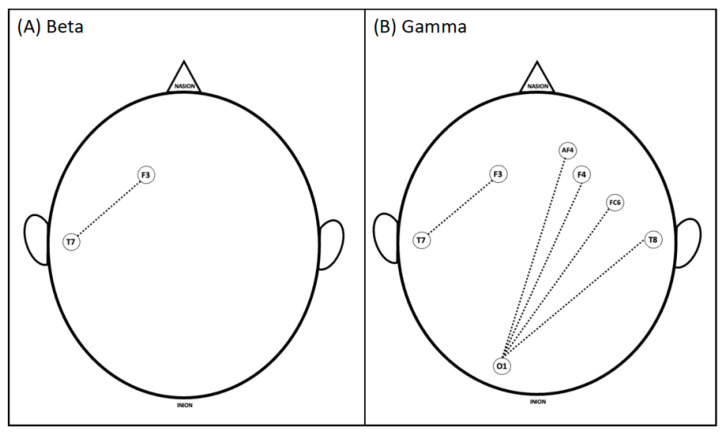
Significant differences in coherence between brain areas of experienced and novice groups during the applied exam for the second case (labeled as answer 2; question-and-answer format) for (**A**) Beta and (**B**) Gamma frequency bands. Dashed lines show where novices had significantly higher coherence.

**Table 1 brainsci-13-01661-t001:** Significant differences in coherence between brain areas of experienced and novice groups in three tasks: reading phase, answering for case 1, and answering for case 2.

Task	Frequency Band	Fellows’ Mean (SD)	Residents’ Mean (SD)	Coherence Channel 1	Coherence Channel 2	*p*-Value
Reading	Theta	0.13 (0.11)	0.52(0.34)	FC5	FC6	0.009
		0.30 (0.35)	0.03 (0.03)	P7	F8	0.047
	Alpha	0.19 (0.15)	0.52 (0.35)	F7	FC6	0.029
		0.20 (0.15)	0.51 (0.360	FC5	FC6	0.038
	Gamma	0.11 (0.16)	0.41 (0.31)	O1	T8	0.033
		0.68 (0.26)	0.32 (0.27)	P8	F8	0.017
Answer 1 Main Question	Theta	0.40 (0.29)	0.14 (0.15)	F7	F8	0.038
	Beta	0.53 (0.25)	0.27 (0.22)	F7	F8	0.047
	Gamma	0.51 (0.27)	0.22 (0.18)	F7	F8	0.028
		0.17 (0.12)	0.50 (0.30)	P7	T8	0.013
Answer 2 Random Question	Beta	0.11 (0.15)	0.40 (0.35)	F3	T7	0.046
	Gamma	0.06 (0.07)	0.44 (0.36)	F3	T7	0.010
		0.07 (0.07)	0.34 (0.32)	O1	T8	0.034
		0.05 (0.04)	0.31 (0.33)	O1	FC6	0.043
		0.15 (0.18)	0.46 (0.35)	O1	F4	0.045
		0.05 (0.03)	0.30 (0.32)	O1	AF4	0.042

**Table 2 brainsci-13-01661-t002:** Results of a linear model with Lasso method for predicting performance.

Variable	Coefficient	Std. Error	*p*-Value
(Intercept)	77.919	9.733	<0.001
Reading, Gamma coherence between F7 and FC5	−18.418	10.17	0.130
Reading, Gamma coherence between P8 and F8	10.723	12.374	0.426
Reading, Gamma coherence between T8 and AF4	4.453	13.226	0.750
Answer 1, Beta coherence between F7 and F8	30.216	13.823	0.081
Answer 1, Gamma coherence between P7 and T8	−17.316	18.468	0.391
Answer 2, Alpha coherence between P7 and P8	−6.21	13.38	0.662
Answer 2, Gamma coherence between F7 and F3	−9.002	14.823	0.570
Answer 2, Gamma coherence between F3 and T7	−4.092	11.078	0.727
Answer 2, Gamma coherence between O1 and F8	−25.169	14.786	0.149
Beta power (PSD), channel F7	−4.811	18.055	0.800

R^2^ = 0.94.

**Table 3 brainsci-13-01661-t003:** Results of a linear model with Lasso method for predicting cognitive load (NASA-TLX).

Variable	Coefficient	Std. Error	*p*-Value
(Intercept)	0.59433	0.03666	<0.001
Answer 1, Theta coherence between F4 and F8	0.16768	0.06632	0.026
Answer 2, Theta coherence between F7 and AF4	−0.25325	0.05181	<0.001
Answer 2, Alpha power (PSD), channel F3	0.12333	0.07644	0.132607

R^2^ = 0.81.

**Table 4 brainsci-13-01661-t004:** Summary of study results and discussion points.

Study Measures	Results	Discussion
Cognitive Load	No difference between groups, *p* = 0.840	Oral examinations require concentration.
Performance	Fellows outperformed novices, *p* < 0.001	Fellows had years of experience and practice taking this type of examination.
Power Spectral Density	No difference between groups	
Coherence Analysis by examination phases		
Reading Clinical Vignette	Fellows showed frontoparietal Theta and Gamma coherence Novices showed frontal Theta and Alpha coherence, and temporal–occipital Gamma coherence	Previous studies show frontal and temporal/parietal Theta phase synchronization for working memory tasks, attentional selection process, and cognitive control [48,49,50,51,52,53,54]. Findings suggest that experienced individuals process information differently due to rehearsal and the ability to retrieve information from memory. Novices rely more on their working memory resources [9,10,11,12,13,14,15,16,17,18,19,20,21].
Answer 1—the main segment of examination based on the clinical vignette	Fellows showed Theta, Beta, and Gamma frontal coherence Novices showed temporoparietal Gamma coherence	Frontal coherence is significantly higher with experience, suggesting fellows engaged in integrating relevant medical knowledge to answer case-specific questions while making clinical decisions. Theta activation reflects cognitive control. Higher evidence of (Beta) active concentration and (Gamma) complex information processing might reflect more efficient problem solving or existing neural pathways [30]. The higher temporal and parietal region activation in novices might reflect the nature of the verbal question-and-answer format of the examination.
Answer 2—questions on a new case without reading preparation. Everything is verbal	Fellows showed no significant coherence Novices showed frontotemporal Beta and Gamma coherence and frontal and temporal–occipital Gamma coherence	Higher Beta and Gamma coherence in novices suggests more cognitive effort in searching for relevant clinical information and integrating visual memory, and coupled with low performance, suggests poorer efficiency of information processing.

## Data Availability

The data presented in this study are available on request from the corresponding author. The data are not publicly available due to human subjects’ privacy concerns.

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
