# Peer review of "Neurocognitive Correlates of Clinical Decision Making: A Pilot Study Using Electroencephalography"

_brainsci, 2023, doi:10.3390/brainsci13121661_

Round 1
Reviewer 1 Report
Comments and Suggestions for Authors
The paper by Serkan Toy et al. is an extremely interesting pilot study on the information that the EEG could provide on the cortical correlates of reasoning strategies in varying degrees of experience and competence. It is a promising starting point and will be of great interest to readers on the topic. I have some suggestions for the Authors:
- I suggest that the Introduction include some background on brain rhythms, on starting assumptions of how they were expected in the two groups and briefly explain what they are and what the importance of PSD and coherence analysis is in this context. Furthermore, the Authors are suggesting that the EEG may in future help to reveal whether the subject's reasoning is valid/is based on sound experience and knowledge? If so, since this is a strong statement with profound implications, I suggest that they elaborate on this concept about the consequences of this possibility (also considering eventual different applications).
- Were there exclusion criteria for participants? For example, were people with traits of anxiety and depression still eligible for your study? Excluded any subjects with drug therapies that could interfere with cognitive performance at some level?
- It is unfortunate that no biometric indices of response to the stress situation were collected to correlate with the EEG picture. For example, is there no information on heart rate or blood pressure?
- Was the experimental protocol conducted at the same day time for all participants? Had anyone consumed coffee and/or smoked before undergoing the evaluation? If so, please evaluate this aspect in your results.
- With regard to the EEG analysis, I would like to ask you to specify what percentage of the epochs were discarded because they were artefactualised, whether the procedure was done automatically or checked manually, what the electrode impedance was during the recording. Also, was the NOTCH filter applied? Was the analysis done with FFT, with Morlet wavelet or with something else? Did you perform an ICA/PCA? Did you only consider gamma up to 45 Hz? If so, what about the high gamma sometimes associated with such situations?
- Were you able to create power spectrograms in the frequency domain to represent the temporal evolution of cortical responses during the examination?
- Is it possible to evaluate a measure such as the Brain Symmetry Index (BSI) to understand, considering the lateralisation you mention in the Introduction, whether there are evolutions between pre- and post-examination sessions or differences at baseline between residents and follows?
- The discussion is certainly very interesting and presents very stimulating observations, but it is scattered and the reader may get lost among all the rhythms and channels examined. I recommend summarising each phase (also in a schematic/tabular form) to make the differences between the two categories examined more immediate for the reader.
Reviewer 2 Report
Comments and Suggestions for Authors
The manuscript focused on an intersting topic and was generally written well. However, the sample size was small (only eight participants per group), and no multiple comparison corrections were performed. These important limitations might lead to unreliable results. Although the authors have claimed this issue in the Limitation section, I still suggest to expand the sample size.
Reviewer 3 Report
Comments and Suggestions for Authors
The study investigates clinical decision-making using electroencephalography (EEG) during a simulated oral exam for anesthesiologists. Participants include novice residents and recently board-certified fellows. While both groups reported similar cognitive load, fellows outperformed novices. EEG analysis revealed significant differences in functional connectivity patterns, particularly in frontal, frontal-parietal, and frontal-temporal areas. These findings offer insights for future studies exploring the cognitive processes influencing clinical reasoning. Please find the detailed comments as follows:
1. Do not use a full stop in the title of the manuscript.
2. There are already some studies done on the similar topic then how this is the pilot study? For example, you can refer the work, “Neurocognitive Models of Medical Decision Making Capacity in Traumatic Brain Injury Across Injury Severity doi: 10.1097/HTR.0000000000000163”. If such work is already done then why there is the need of this work? Authors even not cited this work.
3. Please check the use of the citations. For instance, reference [11] is used twice in the single sentence.
4. The author can add the diagram so that the connections should be clear. Authors have written as “In this study, the participants were fitted with the EMOTIV EPOC+ headset. EPOC+ 157 is a 14-channel wireless headset. The channel names and locations are AF3, F7, F3, FC5, 158 T7, P7, O1, O2, P8, T8, FC6, F4, F8, AF4.” But it is not sufficient. Please check. Also, authors can also cite the source where details can be found.
5. Also, there is no information in the provided link for the webpage, https://emotiv.gitbook.io/emotiv-home/epoc-user-manual. Please check.
6. Authors should identify the limitation or some failed cases of the proposed work and suggest the possible future works based on those identified limitations/failed cases.
7. Section 4 (Discussion) should discuss the other earlier research works and compare them with the proposed work. Otherwise, the superiority of the proposed work could not be established. If tabular comparison can be done, it would be more useful.
8. Paper organization should be provided at the end of Section 1.
9. The significant contributions and novelty of the proposed work are not clear. Please illustrate in a clear manner. The main contributions and novelty of the proposed work should be provided in the Section 1.
10. EMOTIVE has its own paid software. Is there any possibility of doing this experiment with raw EMG connections to save the cost of the experiment?
11. References lack the latest works in this field. Please include recent works.
Comments on the Quality of English LanguageModerate editing of English language required
Round 2
Reviewer 1 Report
Comments and Suggestions for Authors
The Authors properly revised according to the suggestions.
Reviewer 2 Report
Comments and Suggestions for Authors
N/A